# Isoflavones Mediate Dendritogenesis Mainly through Estrogen Receptor α

**DOI:** 10.3390/ijms24109011

**Published:** 2023-05-19

**Authors:** Winda Ariyani, Izuki Amano, Noriyuki Koibuchi

**Affiliations:** 1Metabolic Signal Research Center, Institute for Molecular and Cellular Regulation, Gunma University, 3-39-15 Showa-machi, Maebashi 371-8512, Japan; winda@gunma-u.ac.jp; 2Department of Integrative Physiology, Gunma University Graduate School of Medicine, 3-39-22 Showa-machi, Maebashi 371-8511, Japan; iamano-lj@gunma-u.ac.jp

**Keywords:** genistein, daidzein, S-equol, E2, development, cerebellum, ER, GPER1

## Abstract

The nuclear estrogen receptor (ER) and G-protein-coupled ER (GPER1) play a crucial role during brain development and are involved in dendrite and spine growth as well as synapse formation. Soybean isoflavones, such as genistein, daidzein, and S-equol, a daidzein metabolite, exert their action through ER and GPER1. However, the mechanisms of action of isoflavones on brain development, particularly during dendritogenesis and neuritogenesis, have not yet been extensively studied. We evaluated the effects of isoflavones using mouse primary cerebellar culture, astrocyte-enriched culture, Neuro-2A clonal cells, and co-culture with neurons and astrocytes. Soybean isoflavone-augmented estradiol mediated dendrite arborization in Purkinje cells. Such augmentation was suppressed by co-exposure with ICI 182,780, an antagonist for ERs, or G15, a selective GPER1 antagonist. The knockdown of nuclear ERs or GPER1 also significantly reduced the arborization of dendrites. Particularly, the knockdown of ERα showed the greatest effect. To further examine the specific molecular mechanism, we used Neuro-2A clonal cells. Isoflavones also induced neurite outgrowth of Neuro-2A cells. The knockdown of ERα most strongly reduced isoflavone-induced neurite outgrowth compared with ERβ or GPER1 knockdown. The knockdown of ERα also reduced the mRNA levels of ER-responsive genes (i.e., *Bdnf*, *Camk2b*, *Rbfox3*, *Tubb3*, *Syn1*, *Dlg4*, and *Syp*). Furthermore, isoflavones increased ERα levels, but not ERβ or GPER1 levels, in Neuro-2A cells. The co-culture study of Neuro-2A cells and astrocytes also showed an increase in isoflavone-induced neurite growth, and co-exposure with ICI 182,780 or G15 significantly reduced the effects. In addition, isoflavones increased astrocyte proliferation via ER and GPER1. These results indicate that ERα plays an essential role in isoflavone-induced neuritogenesis. However, GPER1 signaling is also necessary for astrocyte proliferation and astrocyte–neuron communication, which may lead to isoflavone-induced neuritogenesis.

## 1. Introduction

Isoflavones are natural isoflavonoids mainly produced by leguminous plants, such as soybean and red clover [1,2,3]. Genistein (7,4’-dihydroxy-6-methoxyisoflavone) and daidzein (7,4’-dihydroxyisoflavone) are widely known as isoflavone phytoestrogens, and S-equol (4’-methoxy-7-isoflavanol), a major daidzein metabolite, is also ranked in this group [2,4]. These compounds have activities similar to those of 17-β-estradiol (E2). Because of their structural similarity with E2, isoflavones can bind with and activate estrogen receptors (ERs), exerting agonistic or antagonistic effects [1,3]. Furthermore, isoflavones also activate the G-protein-coupled estrogen receptor (GPER1; also known as GPR30), which then initiates several intracellular signal transduction pathways to activate extracellular signal-regulated kinase 1/2 (ERK1/2) and/or Akt-mediated pathways [5,6,7,8]. Such various actions of isoflavones indicate that they act through several signaling pathways. Furthermore, isoflavones can also affect other pathways. For example, they bind to membrane receptors such as peroxisome proliferator-activated receptor-γ [1,9] and without binding to a specific receptor, they function as antioxidants [1,4,10]. Because of these multiple pathways, they modulate the transcription of estrogen-regulated genes and chromatin remodeling, the activity of protein kinases, and the action of various transcription factors [11,12]. Thus, the molecular mechanism of action of isoflavones has not yet been fully understood, although the major pathway may be mediated through ER or GPER1.

Estrogens, particularly E2, play an essential role in the development and functional maintenance of the central nervous system, particularly through binding with ERs, including the cerebellum, which consists of various cell types, such as Purkinje cells, granule cells, basket cells, stellate cells, astrocytes, and many more, integrated into a cytoarchitecture array of stripes and zones [13]. Isoflavones may affect Purkinje cell dendrite and spine growth and synapse formation. Furthermore, they also affect the proliferation, migration, and differentiation of other neurons and glial cells and neuritogenesis and synaptogenesis in these cells [6,14,15,16,17]. Exposure to isoflavones prevents low-potassium-dependent apoptosis by preventing the impairment of glucose oxidation and mitochondrial coupling, reducing cytochrome c release and preventing both impairments of the adenine nucleotide translocator and opening of the mitochondrial permeability transition pore in cerebellar granule cells [18]. Therefore, exposure to isoflavones during development affects the neurobehavior and cognitive function of the offspring [19,20,21,22,23,24]. 

Despite the various actions of isoflavones, the underlying mechanism of each isoflavone compound in the ER or GPER1 of specific subsets of cells remains poorly understood. Our previous study showed that isoflavones activate GPER1 in astrocytes and enhance the migration [6]. Additionally, we found that S-equol, a daidzein metabolite, activates ER and GPER1 to augment thyroid hormone (TH)-mediated dendrite arborization in Purkinje cells; however, the mechanism was unknown [5]. Therefore, we aimed to determine the effects of isoflavones during dendritogenesis and neuritogenesis using primary cerebellar culture and a neuronal-derived clonal cell line, Neuro-2A. We also examined the involvement of astrocytes during neuritogenesis. We found that isoflavones induced different signaling pathways for different subsets of cells. Each isoflavone compound also has different selectivity or affinity to activate ER or GPER1. Exposure to isoflavones exclusively activates and modulates ERα, which directly affects neuritogenesis.

## 2. Results

### 2.1. Isoflavones Enhanced Cell Number and Dendrite Arborization in Purkinje Cells in the Primary Culture of Mouse Cerebellar Cells

Our previous study showed that S-equol augmented the TH-mediated dendrite arborization in Purkinje cells in the primary cerebellar culture, and co-exposure with ER or GPER1 inhibitors significantly reduced the effect [5]. However, whether TH plays some roles in this system through TH receptor (TR)-dependent, TR-independent, or crosstalk of ER–TR signaling pathways cannot be excluded. To elucidate the effects of isoflavones on the dendrite arborization in Purkinje cells, we observed the effects of isoflavones in the primary cerebellar cultures without TH supplementation. The culture cells were maintained in a serum-free medium, and dendrite arborization in Purkinje cells was induced by adding E2 (10 nM) and/or genistein, daidzein, or S-equol (10 nM) (Figure 1A) to the culture medium. In addition to Purkinje cells, these cultures contained granule cells, astrocytes, and interneurons. The cells were cultured for 17 days and then fixed and immunostained with an anti-Calbindin D-28K antibody to visualize dendrite arborization in Purkinje cells; the dendritic area in Purkinje cells was measured and the total number of cells was counted. Representative photomicrographs after E2 and/or isoflavone exposure are shown in Figure 1B. We found that isoflavones enhanced E2-mediated dendrite arborization in Purkinje cells, and no significant difference was observed between the control and isoflavone-only groups (Figure 1C). Furthermore, isoflavones also increased the total number of cells, particularly Purkinje cells, in the primary cerebellar cultures (Figure 1D,E). These results indicate that isoflavones with E2 affected both neurons and glial cells and induced dendritogenesis and cell proliferation during cerebellar development. However, the molecular mechanism underlying these effects should be addressed further.

### 2.2. GPER1 and ER Are Required for Isoflavone-Enhanced E2-Mediated Dendrite Arborization in Purkinje Cells

Isoflavones exert their action mainly by binding to either ERα or ERβ of the nuclear isoforms of ER, which then bind with an estrogen-responsive element located in the promoter region of the target genes to regulate gene transcription. In addition to nuclear ERs, isoflavones also bind to GPER1 and activate several intracellular signaling pathways [1,2,3,6,14]. Therefore, to examine the involvement of ER and GPER1 in dendritic outgrowth in Purkinje cells, we exposed the cells to isoflavones with G15, a GPER1 antagonist, or ICI 182,780, an ER antagonist. Co-exposure with 10 nM ICI 182,780 or G15, suppressed the isoflavone-augmented dendrite arborization in Purkinje cells (Figure 2A,B). We also observed the involvement of ER in dendritic outgrowth in Purkinje cells; we used siRNAs against ERα, ERβ, or GPER1 to knockdown their RNA expression in a primary culture of cerebellar cells. The mRNA levels after siRNA treatment were confirmed to be significantly suppressed (Appendix A). We also performed double knockdown of both ER mRNAs. DsiRNA as a control did not significantly affect dendrite arborization in Purkinje cells. In contrast, the knockdown of ERα and/or ERβ significantly reduced dendrite arborization in Purkinje cells (Figure 2C–F). Furthermore, the knockdown of GPER1 also suppressed daidzein- and S-equol-augmented dendrite arborization in Purkinje cells. However, no significant difference was observed in the genistein group (Figure 2G). These results indicate that ER and GPER1 are essential for dendrite arborization in Purkinje cells. However, it should be noted that we were unable to confirm whether the knockdown of ER or GPER1 directly affected Purkinje cells or that this effect was mediated by other cellular subsets.

### 2.3. ERα Mediated Isoflavone-Induced Neurite Outgrowth in Neuro-2A Cells

As shown in Figure 1 and Figure 2, the primary cerebellar cultures contain several subsets of cells. The action of isoflavones on the primary cerebellar culture may be mediated by their direct effects on Purkinje cells or via cell–cell interactions. Hence, to clarify the mechanism of action of isoflavones during development, we used a Neuro-2A cell differentiation model. The serum concentration in the culture medium was reduced to 1% to induce the differentiation of Neuro-2A cells. Anti-β-tubulin III for mature neurons and anti-doublecortin (DCX) for immature neurons or a neurogenesis marker were used in the Neuro-2A cells immunostained as neuronal markers [5,25]. E2 or isoflavones (10 nM) enhanced neurite outgrowth in Neuro-2A cells (Figure 3A,B). Neurite lengths were quantified using ImageJ Fiji (NIH). Using this system, we further elucidated the involvement of ER and GPER1 during isoflavone-induced neuritogenesis. Co-exposure with ICI 182,780 10 nM significantly suppressed isoflavone-induced neuritogenesis in Neuro-2A cells (Figure 3C). In contrast, co-exposure with the GPER1 inhibitor (G15 10 nM) only affected S-equol- and E2-induced neuritogenesis in Neuro-2A cells (Figure 3D). We also examined the mechanism using siRNA against ERα, ERβ, or GPER1 to knockdown RNA expression in Neuro-2A cells. A significant suppression in the mRNA levels after siRNA treatment was confirmed (Appendix A). Surprisingly, we found that only the knockdown of ERα significantly reduced isoflavone-induced neurite outgrowth in Neuro-2A cells (Figure 3E,G), whereas the knockdown of ERβ enhanced neurite outgrowth in the control, genistein, and E2 groups (Figure 3F). Furthermore, the knockdown of GPER1 reduced S-equol- and E2-induced neuritogenesis (Figure 3H). These results demonstrate that ERα is involved in the neuritogenesis in Neuro-2A cells that was induced by isoflavones.

Furthermore, we manually traced each differentiated Neuro-2A cell to quantify the morphology of neurites using Sholl analysis. We examined the number of intersections between neurites, and each concentric circle was measured and analyzed (Figure 4A). Exposure to isoflavones increased neurite outgrowth in Neuro-2A cells, including the branching and neurite length (Figure 4B). We further examined the involvement of ER and GPER1 during neuritogenesis using siRNA. We found that the knockdown of ERα significantly reduced isoflavone-induced neurite outgrowth in Neuro-2A cells, whereas the knockdown of GPER1 reduced the amount of branching without affecting the neurite length (Figure 4C–F). In contrast, the knockdown of both ERs enhanced isoflavone-induced neurite branching from the soma of Neuro-2A cells. These results may be because of the effects of isoflavone binding via GPER1. Furthermore, we also found that exposure to isoflavones significantly increased ERα expression but not ERβ or GPER1 (Figure 4G–J). Thus, we continued to examine the mRNA levels of ERα-responsive genes related to neuritogenesis in Neuro-2A cells via quantitative real-time polymerase chain reaction (qRT-PCR). The mRNA levels of *Bdnf*, *Camk2b*, *Rbfox3*, *Tubb3*, *Syn1*, *Dlg4*, and *Syp* were significantly decreased after the knockdown of ERα in Neuro-2A cells (Figure 5A–H). These results indicate that isoflavones exert their action though the activation and modulation of ERα, which led to neuritogenesis in the neurons. 

### 2.4. Neuron–Glia Interaction Enhanced Isoflavone-Mediated Neuritogenesis

Our previous studies showed that isoflavones affect both neurons and astrocytes through different signaling pathways [5,6]. Isoflavones also affect F-actin rearrangement, which may change cell adhesion, morphology, and movement, inducing the interaction and communication between cells. During cerebellar development, cell-to-cell communication between neurons and other cells, including astrocytes, plays an important role and greatly influences the maturation of these cells [26,27,28,29,30,31,32]. We examined the effects of neuron–glia interaction on the action of isoflavones via a co-culture study between Neuro-2A cells and astrocytes. We performed astrocyte-enriched culture until 70–80% confluence0 and then added Neuro-2A cells. The serum was reduced to induce the differentiation of Neuro-2A cells, and cultures were maintained for 3–5 days. Then, Neuro-2A cells were immunostained with β-tubulin III (blue), F-actin (green), and S100B (astrocytes marker) (red) to examine the neurite outgrowth. We found that the co-culture of Neuro-2A and astrocytes enhanced isoflavone-induced neurite outgrowth, and these effects were significantly reduced by co-exposure with ICI 182,780 or G15 (Figure 6A–D). In Figure 1E, we found that isoflavones increased the total number of cells in the primary cerebellar cultures. Therefore, we also examined the effects of isoflavones on the proliferation of astrocytes. The cerebellar astrocyte-enriched culture was maintained for 1–3 days after the addition of isoflavones and ICI 182,780 or G15, followed by immunostaining with Ki-67 as a proliferation marker. We found that exposure to isoflavones significantly increased the number of Ki-67-positive cells (Figure 6E,F). Furthermore, co-exposure with ICI 182,780 or G15, as an ER or GPER1 antagonist, suppressed the isoflavone-induced increase in the number of Ki-67-positive cells (Figure 6G,H). These results indicate that isoflavones exert their action through both ER and GPER1. However, each isoflavone compound has different affinity and works differently in each subset of cells. 

## 3. Discussion

In our study, we demonstrated that isoflavones affect dendritogenesis and neuritogenesis through the activation and modulation of ERα. We found that, in the primary cerebellar cultures, both ER and GPER1 affected the dendrite arborization and total number of Purkinje cells. We cannot clarify the specific mechanism of such an alteration because the action may be exerted through other subsets of cells included in the culture. Thus, we decided to identify the specific action of isoflavones via ERα to enhance neurite outgrowth in Neuro-2A cells. In this cell type, isoflavones also increased ERα levels, and the knockdown of ERα reduced the mRNA levels of ER-responsive genes (i.e., *Bdnf*, *Camk2b*, *Rbfox3*, *Tubb3*, *Syn1*, *Dlg4*, and *Syp*), whereas no significant difference was observed in these mRNA levels after the knockdown of ERβ or GPER1. However, in the case of S-equol and E2, co-exposure with G15 or the knockdown of GPER1 significantly reduced S-equol- and E2-induced neurite outgrowth. These results may be because of the higher binding affinity of S-equol and E2 to GPER1. Isoflavones also increased neurite outgrowth in the co-culture of Neuro-2A with a cerebellar astrocyte-enriched culture and enhanced the proliferation ability of astrocytes through both ER- and GPER1-mediated signaling pathways. Our results demonstrate a novel mechanism of action of isoflavones to promote dendritogenesis and neuritogenesis by activating and modulating ERα. Isoflavones also increased the communication between astrocytes and neurons and the proliferation of astrocytes through both ER and GPER1, which increased neurite outgrowth.

The structure of the cerebellum is deeply integrated into primary loops with the cerebral cortex, brainstem, and spinal cord and plays an essential role in integrating and processing motor and sensory information, such as motor coordination, balance, posture, the precise timing of movements, and motor learning [13,33,34]. Moreover, recent brain imaging studies have shown the cerebellar contribution in cognitive functions, such as attention, language, working memory, emotion, and visuospatial navigation [35]. The scientific evidence regarding the effects of isoflavones on the cerebellum has been well documented. Exposure to isoflavones prevents low-potassium-dependent apoptosis by preventing the impairment of glucose oxidation and mitochondrial coupling, reducing cytochrome c release, and preventing both impairments in the adenine nucleotide translocator and opening of the mitochondrial permeability transition pore in cerebellar granules cells [18]. Furthermore, exposure to isoflavones increased dendrite arborization in Purkinje cells and cell proliferation and migration and stimulated Ca^2+^ influxes via Na^+^/Ca^2+^ exchange in cerebellar astrocytes [5,6,36]. Genistein exposure also blocked the activation of mGluR1, the alternative splicing of canonical transient receptor potential (TRPC3) mRNA transcript (designated TRPC3c) in cerebellar Purkinje cells [37]. These results show that isoflavones affect various subsets of cells in the cerebellum.

Human brain development starts about two weeks after conception and continues into young adulthood [38]. It is difficult to study the brain development of the human embryo due to the ethical and practical constraints on research with humans. Therefore, researchers have relied on alternative models, such as animals. The mouse model is the most commonly used model to study brain development due to the relative ease of genetic manipulations [39]. However, due to the complicated process during brain development involving various cell types and signaling pathways, researchers still combine in vitro and in vivo animal models to examine brain development. Brain development that occurs during the prenatal months is largely under genetic control. However, environmental factors can also play an important role, i.e., a lack of nutrition (e.g., folic acid) and toxins (e.g., alcohol) can influence the developing brain. Thus, brain development is defined by gene–environment interactions, even during the fetal period [38]. E2 and ERs exert potent and wide-ranging effects on the development and functional maintenance of the brain [40]. E2 actions may have life-long impacts by influencing proliferation, migration, and survival of glia and neurons, axonal projection, dendritic branching, and synaptic patterning in brain function [40,41]. In addition, the presence of dietary isoflavones as a phytoestrogen in the developing brain may enhance or suppress E2 action through additive or competitive binding with ERs or other receptors. In the present study, we used the CE-2 standard rodent diet from CLEA Japan, Inc., which contains soybean and soybean oil. Thus, the possibility that the prenatal exposure of isoflavones might affect the results of the present study cannot be excluded. Nevertheless, we showed the involvement of ERs and GPER on dendritogenesis using an early postnatal mouse brain, indicating the possible effects of soy isoflavones on rodent brain development during the early postnatal period, which may mimic the late gestational period in humans. 

Exposure to isoflavones during brain development also affects axonal and dendritic growth through various signaling pathways. Exposure to isoflavone-enriched fractions that contain genistein and daidzein induced cell differentiation and neuritogenesis in human cortical neuronal cells-1A [42]. Our previous study showed that S-equol exposure increased dendrite arborization in Purkinje cells in primary cerebellar cultures and neurite outgrowth in Neuro-2A cells. These effects were induced through both ERs and GPER1 [5]. Daidzein exposure enhanced neuritogenesis in the primary cultures of rat dorsal root ganglia via the activation of Src, protein kinase C delta, and mitogen-activated protein kinase/extracellular signal-regulated kinases (MEK/ERKs) [43]. Genistein exposure activated GPER1, which induced acetylcholinesterase via a cAMP response element binding site, leading to differentiation and neurite outgrowth in PC12 neuronal cells [44]. In addition, genistein stimulates the nerve growth factor-induced neurite outgrowth via the activation of Na^+^/K^+^/2Cl^−^ cotransporter isoform 1 also in PC12 neuronal cells [45]. These results indicate that each isoflavone compound differentially activates the various signaling pathways necessary for normal cerebellar development.

Interactions between neurons, astrocytes, and microglia play an essential role during development, mediating metabolic processes, supplying nutrients, and improving brain metabolism and function. During development, the surrounding environment and cell-to-cell communication between neurons, astrocytes, microglia, endothelial cells, and oligodendrocytes greatly influence their maturation [26,27,28,29,30,31,32]. Because of this close relationship, slight environmental variations can cause changes in the phenotype of cells, gene expression, and intracellular dynamics of Ca^2+^ in the brain [32]. Furthermore, isoflavones have been reported to affect neurons, astrocytes, and microglia during development and in adulthood. Exposure to isoflavones inhibited lipopolysaccharide-induced microglial activation, leading to protective effects on dopaminergic neurons in rat mesencephalic neuron–glia cultures [46,47]. The equol-treated microglia culture medium induced cell proliferation and neuritogenesis in the Neuro-2A neuronal cell line [48]. Therefore, exposure to isoflavones during development may affect the characteristics and morphology of neurons, astrocytes, and microglia. 

Despite the broad spectrum of health benefits of isoflavones, there is growing evidence from human dietary and epidemiological studies that the role of isoflavones in human health is questionable. Some studies have demonstrated the effects of isoflavones as dietary supplements and alternative hormone replacement therapy, and other studies have failed to prove the favorable effects of isoflavones. Moreover, the dose of isoflavones is also crucial to examine their effects on human health because of the biphasic dose–response relationship affecting multiple endpoints. Soybeans have been consumed throughout human evolution, and isoflavones can be effectively metabolized in humans. The half-lives of plasma genistein, daidzein, and S-equol are 8.36, 5.79, and 7 h, respectively [49,50]. The total plasma concentration of isoflavones in the Asian population consuming a traditional diet, including soy-based food, is in a range of 525–775 nM. In contrast, in European countries, the total plasma concentration of isoflavones was found to be <10 nM in individuals consuming a nonvegetarian diet and 79–148 nM in those consuming vegetarian and vegan diets [51]. The plasma levels of genistein and daidzein in patients with prostate cancer reached 1–10 µM, 2–8 h after the intake of two slices of standard soy or soy-almond bread [52], and reached 1–10 µM 4–7 h after the intake of 50 g of *kinako* (roasted soybean powder) in human subjects [53]. Because this was an in vitro study, the results cannot be compared with those from in vivo conditions. However, we highlight the novel possibility that isoflavones enhance dendritogenesis and neuritogenesis, indicating that they can be a useful supplementary compound during brain development or in the injured brain.

## 4. Materials and Methods

### 4.1. Materials

Genistein, daidzein, E2, and ICI 182,780 were purchased from Sigma (St. Louis, MO, USA). S-equol and G-15 were purchased from Cayman Chemical (Ann Arbor, MI, USA). The purity of all chemicals was above 98%. 

### 4.2. Primary Culture of Mouse Cerebellar Cells

A primary culture of cerebellar cells was prepared as described previously [5,25,54] with slight modifications. Pregnant mice (C57BL/6) were purchased from Japan SLC (Hamamatsu, Japan) and fed with CE-2 standard rodent diet from CLEA Japan, Inc. At postnatal day 1, mouse cerebellar tissue was digested with papain dilution buffer (0.2 units/mL papain (Worthington, Lakewood, NJ, USA) in phosphate-buffered saline (PBS) containing 0.2 mg/mL l-cysteine, 0.2 mg/mL bovine serum albumin (Intergen Company, Purchase, New York, NY, USA), 5 mg/mL glucose, and 0.02 mg/mL DNase I (Sigma)) or underwent continued shaking at 37 °C for 25 min. Cells were resuspended in a cerebellar culture medium (Dulbecco’s Modified Eagle’s Medium/F12 (DMEM/F12) (Fujifilm Wako Pure Chemical Co., Ltd., Kanagawa, Japan) containing 1% penicillin–streptomycin (Wako), 3.9 mM glutamine (Wako), 2.1 mg/mL glucose (Wako), 30 nM sodium selenite (Sigma), 20 mg/mL insulin (Sigma), and 200 mg/mL transferrin (Sigma)). Dissociated cell suspensions were counted, and 3 × 10^5^ cells/0.3 mL cerebellar cells were plated in poly-l-lysine-coated 8 mm diameter wells of chamber slides (Lab-Tek; Nunc International, Rochester, New York, NY, USA). E2, genistein, daidzein (Sigma), or S-equol (Cayman) at concentrations of 10 nM were added to the culture medium 16–24 h after plating. One-half of the cerebellar culture medium was replaced with fresh medium every 3 days and incubated in a 5% CO_2_ incubator at 37 °C.

After 17 days of culture, mixed cerebellar cells were fixed with 4% paraformaldehyde (PFA). The cells were blocked with 5% fetal bovine serum (FBS) in PBS and immunostained with a Purkinje cell marker, which is mouse monoclonal anti-calbindin-D-28K antibody (1:200; Sigma; catalog number: C9848), followed by secondary antibody, donkey anti-mouse IgG (H + L) Alexa Fluor^®^ 594 conjugate (1:200; Thermo-Fisher Scientific, Inc, Waltham, MA, USA; catalog number: A11005). Cell nuclei were stained with DAPI. Purkinje cells were examined under a laser confocal scanning microscope with a 40X objective lens, pinhole 1 Airy Unit (AU), image size 512 × 512 pixels, and bit depth 8 bit (Zeiss LSM 880, Carl Zeiss Microscopy GmbH, Jena, Germany). For each experiment, 20–50 Purkinje cells were randomly selected to quantify dendrite arborization. The dendritic areas of Purkinje cells were manually traced with Fiji ImageJ (NIH, Bethesda, MD, USA) Freehand selection tools, and the total area of dendrite arborization was measured. 

### 4.3. Primary Culture of Mouse Cerebellar Astrocytes

A primary culture of mouse cerebellar astrocytes was prepared as described previously [5,6,25,55] with slight modifications. Briefly, pups were euthanized under isoflurane anesthesia. The cerebellum was dissected and then digested with Hank’s balanced salt solution (Wako) containing 2.5% trypsin (Wako). Mixed cerebellar tissue was incubated at 37 °C for 30 min with continuous shaking and then centrifuged at 3500× *g* rpm for 15 min at 4 °C. The cells were resuspended in an astrocyte culture medium (high-glucose DMEM (Wako), 10% heat-inactivated FBS, and 1% penicillin/streptomycin), then counted and 10–15 × 10^6^ cells were plated on collagen-I-coated 10 cm dishes (Iwaki Science Products Dept., AGC Techno Glass Co., Ltd., Shizuoka, Japan) and incubated at 37 °C in a CO_2_ incubator. The astrocyte culture medium was replaced with PBS on day 3 in vitro (DIV3). Oligodendrocyte precursor cells were removed by shaking the dishes for 2–5 min. The culture medium was replaced with fresh astrocyte culture medium. Astrocytes were harvested on DIV7 and then plated on collagen-I-coated 6- or 24-well dishes (Iwaki) to use for further studies. A cell proliferation assay was performed using the plated DIV7 cerebellar astrocytes at a density of 1 × 10^5^ cells/mL per well on collagen-I-coated 24-well dishes, and then the mixture was incubated in the presence of E2, genistein, daidzein, or S-equol (10 nM) with or without ICI 182,780 or G15 (10 nM). The cells were fixed with 4% PFA, followed by blocking with 5% FBS in PBS for 30 min. Astrocytes were then immunostained with the Ki-67 monoclonal antibody (Ki-67), Alexa Fluor™ 488 (1:200; Thermo-Fisher Scientific, Inc.; catalog number: MA544124). DAPI was used to stain the cell nuclei. Cerebellar astrocytes were observed under a laser confocal scanning microscope with a 20X objective lens, pinhole 1 AU, image size 512 × 512 pixels, and bit depth 8 bit (Zeiss LSM 880). ImageJ Fiji (NIH) was used to measure the cell number.

### 4.4. Mouse Neuro-2A and Co-Culture Studies

Mouse neuroblastoma-derived clonal cells, Neuro-2A cells (Japanese Cancer Research Resources Bank cell bank (National Institutes of Biomedical Innovation, Health, and Nutrition, Tokyo, Japan)), were cultured in DMEM (Wako) supplemented with 10% FBS and 100 U/mL penicillin and 100 µg/mL streptomycin (Wako) with 5% CO_2_ at 37 °C. The FBS was depleted of hormones using AGXI-8 resin (Bio-Rad, Hercules, CA, USA) through constant mixing and ultrafiltration. Then, 1 × 10^5^ Neuro-2A cells/mL per well were plated in poly-l-lysine-coated 6- or 24-well plates and cultured in DMEM + 10% FBS. The culture medium was changed to DMEM + 1% FBS on the next day with or without E2, genistein, daidzein, or S-equol (10 nM) to trigger differentiation [5]. Then, the cells were incubated at 37 °C in a CO_2_ incubator for 1–3 days and then used for RT-qPCR and immunocytochemistry. Neuro-2A cells were fixed with 4% PFA and then blocked with 5% FBS in PBS for 30 min. The cells were then immunostained with a neuronal marker, mouse anti-β-tubulin III (1:200; Sigma; catalog number: T8578) and rabbit anti-doublecortin (C–terminal) (1:200; Sigma; catalog number: D9818) antibodies, followed by secondary antibody donkey anti-mouse Alexa Fluor^®^ 405 (1:200; Thermo-Fisher Scientific, Inc.; catalog number: A31553) and donkey anti-rabbit Alexa Fluor^®^ 594 conjugate (1:200; Thermo-Fisher Scientific, Inc.; catalog number: A32740). F-actin was stained with CytoPainter Phalloidin-iFluor 488 reagent (Abcam; catalog number: ab176753). Neuro-2A cells were observed under a laser confocal scanning microscope with a 40× objective lens, pinhole 1 AU, image size 512 × 512 pixels, and bit depth 8 bit (Zeiss LSM 880). The cells were randomly selected to quantify the neurite length and perform Sholl analysis. Neurite length was measured by tracing the neurite outline of the cells using Fiji ImageJ (NIH). Sholl analysis was performed by manually tracing each cell using Adobe Illustrator (Adobe Inc., San Jose, CA, USA). The overlay of the transparent layer of the Neuro-2A cells was used to trace the neurite outgrowth, followed by Sholl analysis using Fiji ImageJ (NIH). Intersections between neurites and each concentric circle were measured and analyzed. 

Co-culture assays were performed as described previously [25] with slight modifications. Co-culture studies were conducted by adding Neuro-2A cells with a density of 1000 or 10,000 cells per well onto monolayers of DIV7 cerebellar astrocyte-enriched culture (70–80% confluence) in 24- or 6-well plates. Co-culture was performed for 3 days, unless otherwise specified. E2, genistein, daidzein, or S-equol (10 nM) was added to the culture medium with or without ICI 182,780 or G15 (10 nM). The cells were fixed with 4% PFA, followed by blocking with 5% FBS in PBS for 30 min. Then, the Neuro-2A cells were immunostained with neuronal marker mouse anti-β-tubulin III antibody (1:200) followed by the secondary antibody donkey anti-mouse Alexa Fluor^®^ 405 (1:200). F-actin was stained with CytoPainter Phalloidin-iFluor 594 reagent (Abcam). DAPI was used to stain the cell nuclei. The Neuro-2A cells were observed under a laser confocal scanning microscope with a 40X objective lens, pinhole 1 AU, image size 512 × 512 pixels, and bit depth 8 bit (Zeiss LSM 880, Carl Zeiss Microscopy GmbH). ImageJ Fiji (NIH) was used to measure the neurite length. 

### 4.5. RNA Interference Assay

An RNA interference assay was performed as described previously [5,6,25,55] with slight modifications. The siRNAs were transfected into the primary cultured cerebellar cells or Neuro-2A cells using lipofectamine RNAiMAX reagent (Thermo-Fisher Scientific) according to the manufacturer’s protocol. The siRNA sequences against ERα, ERβ, GPER1, or negative control DsiRNA (Integrated DNA Technologies, Inc., Coralville, IA, USA) used in this study are listed in Appendix A. Briefly, 1 nM of DsiRNA, ERα, ERβ, or GPER1 siRNA in siRNA lipid complexes was incubated for 20 min at room temperature. Primary cultured cerebellar cells or Neuro-2A cells were cultured until 80% confluences in 24-, 96-well, or 8 mm diameter wells of chamber slides in poly-l-lysine-coated dishes. Then, siRNA lipid complexes were added dropwise to the well. The qRT-PCR was performed to verify the efficacy of siRNA knockdown (Appendix A). The *SuperPrep* cell lysis and RT kits for qPCR reagent (TOYOBO Bio-Technology, Japan) were used to extract RNA, followed by qRT-PCR using THUNDERBIRD SYBR qPCR Mix (TOYOBO) according to manufacturer’s instructions and a StepOne RT-PCR System (Thermo-Fisher Scientific). The primers are listed in Appendix A. qRT-PCR was performed as follows: denaturation at 95 °C for 20 s; amplification at 95 °C for 3 s; and 60 °C for 30 s (40 cycles). Data are presented as the mean ± standard error of the mean. More than three independent experiments were performed, using independent RNA preparations to ensure the consistency of the results. Glyceraldehyde-3-phosphate dehydrogenase (*Gapdh*) was used to normalize mRNA levels. The results were consistent between each experiment. A representative result from one experiment is shown.

### 4.6. Western Blotting

Western blotting was performed as described previously [6,25,55] with slight modifications. Neuro-2A cells were homogenized in Radioimmunoprecipitation assay (RIPA) buffer (Cell Signaling) and protease inhibitors (Complete; Roche, IN, USA). The Bradford protein assay (Bio-Rad) was used to measure the protein concentration according to the manufacturer’s instructions. After boiling for 5 min, protein samples (5 µg) were subjected to 5–20% SDS–polyacrylamide Supersep Ace (Wako) gel electrophoresis, and the separated products were transferred to nitrocellulose membranes. Membranes were blocked with 5% nonfat dry milk in Tris-buffered saline containing 0.1% Tween 20, followed by overnight incubation with the appropriate diluted primary antibodies for ERα (1:1000; MerckMilipore, CA, USA; catalog number: 06935), ERβ (1:1000; GeneTex, Inc, CA, USA; catalog number: GTX70174), GPER1 (1:1000; Alomone Labs, JBP, Israel; catalog number: AER050), and GAPDH (1:1000; Proteintech, IL, USA; catalog number: 600041Ig). After washing with Tris-buffered saline containing 0.1% Tween 20, the membranes were incubated with horseradish peroxidase-conjugated anti-rabbit or anti-mouse IgG secondary antibody (1:3000; Cell Signaling; catalog number: 7074) for 1 h at room temperature and detected using an ECL detection system (Wako). GAPDH was once again used as the loading control.

### 4.7. Software and Statistical Analyses

Multi-panel figures were assembled in Adobe Illustrator (Adobe). Data were analyzed using GraphPad Prism (version 9.5.0; GraphPad Software, San Diego, CA, USA, www.graphpad.com, accessed on 18 May 2023) and are presented as the mean ± standard error of the mean. Student’s unpaired *t*-test (for two-group comparisons) and one-way or two-way analysis of variance followed by post hoc Tukey multiple comparison test were performed for data analyses. Differences with *p*-values < 0.05 were considered statistically significant.

## 5. Conclusions

In conclusion, the evidence presented here indicates that isoflavones, such as genistein, daidzein, and S-equol, play a fundamental role in cerebellar development. Isoflavones can enhance dendritogenesis and neuritogenesis via the ERα signaling pathway. Furthermore, isoflavones also activate the GPER1 signaling pathway to increase astrocyte proliferation and astrocyte–neuron interaction, affecting neurite outgrowth. The positive or negative neurodevelopmental effects of isoflavones depend on the timing, dose, and duration of exposure/treatment. Timing plays an essential role because of the nature of brain development, and the dose of isoflavones is also important because of their biphasic effects. Despite an increasing number of studies, data on the biological effects of isoflavones and their impact on brain development remain limited. Isoflavones are promising substances that may provide new insights on physiological regulations and therapeutic interventions during brain development.

## Figures and Tables

**Figure 1 ijms-24-09011-f001:**
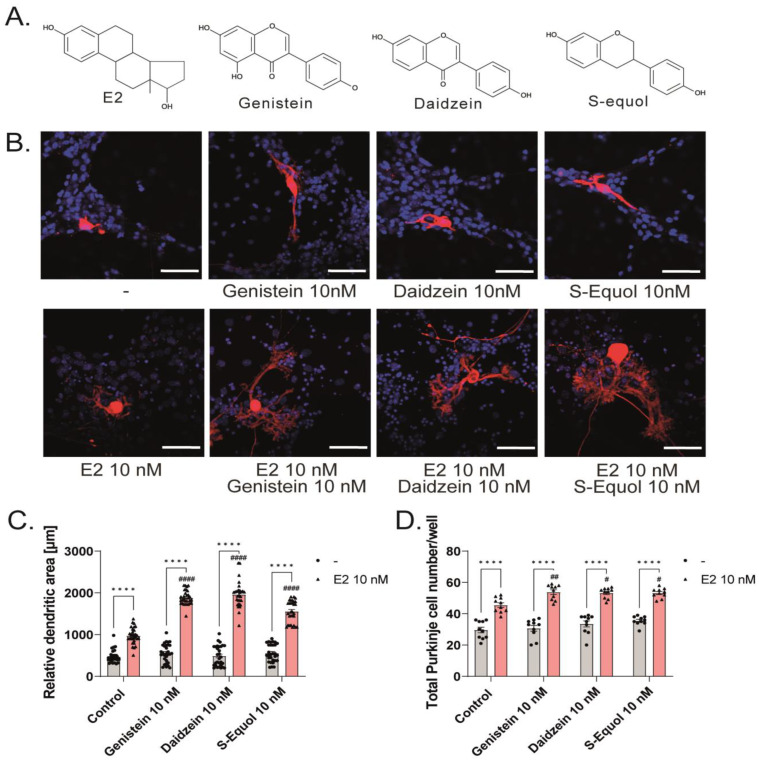
Effects of isoflavones in the primary cerebellar cultures. Cerebellar cells were cultured for 17 days, followed by an immunocytochemistry analysis of Purkinje cells with Calbindin D-28K (red) and 4′,6-diamidino-2-phenylindole (DAPI) (blue). (**A**) Chemicals structures of isoflavones and E2. (**B**) Representative photomicrographs showing the effects of isoflavones on the morphology of Purkinje cells. (**C**) Quantitative analysis of the effects of isoflavones on the dendritic areas in Purkinje cells (n = 30). Changes in the total number of Purkinje cells (n = 10) (**D**) and total cells (n = 30) (**E**) following treatment with isoflavones. Bars indicate 50 μm. Data are shown as the mean ± standard error of the mean and represent at least three independent experiments. ^####^
*p*  <  0.0001, ^##^
*p*  <  0.01, and ^#^
*p*  <  0.05 indicates statistical significance according to two-way or one-way analysis of variance, followed by the post hoc Tukey test compared to E2 10 nM. **** *p* < 0.0001 and ** *p* < 0.01 indicate statistical significance according to two-way or one-way analysis of variance, followed by the post hoc Tukey test.

**Figure 2 ijms-24-09011-f002:**
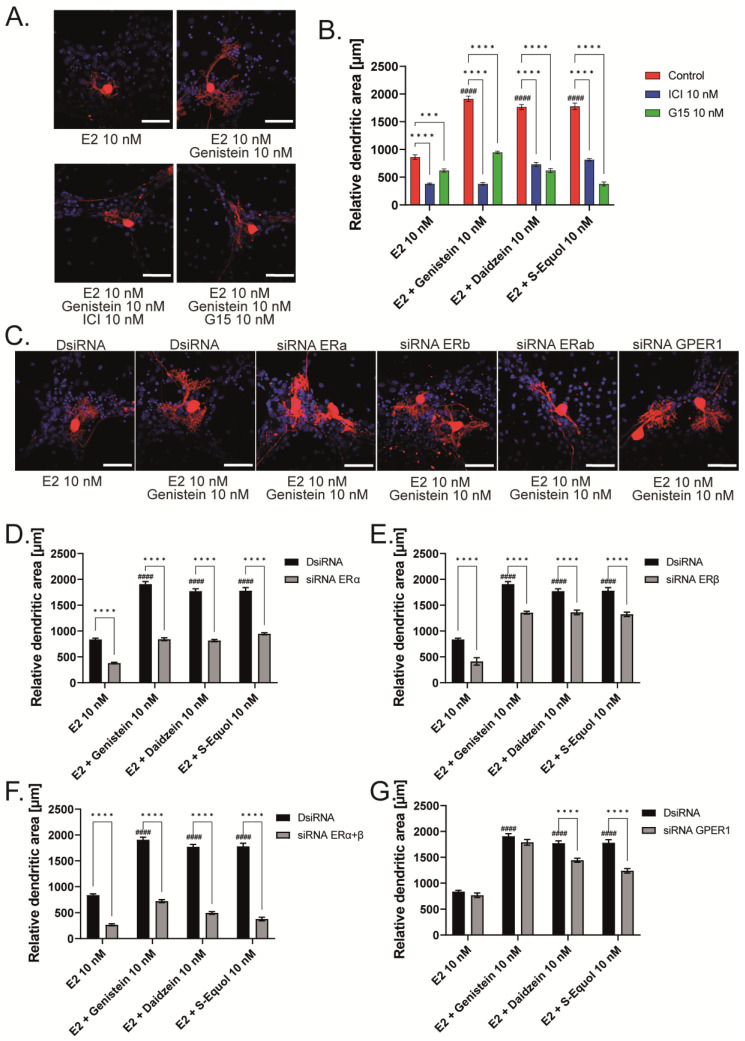
ER and GPER1 are essential for isoflavone-augmented dendrite arborization in Purkinje cells. Cerebellar cells were cultured for 17 days, followed by an immunocytochemistry analysis of Purkinje cells with Calbindin D-28K (red) and DAPI (blue). (**A**) Representative photomicrographs showing the co-exposure effects of genistein with ICI 182,780 or G15 on the morphology of Purkinje cells. (**B**) Changes in the dendritic areas in Purkinje cells following S-equol, G15, and/or ICI 182,780 treatment (n = 15). (**C**) Representative photomicrographs showing the effects of genistein on the morphology of Purkinje cells after ERα, ERβ, or GPER1 knockdown. Quantitative analysis of the effects of isoflavones after exposure to ERα siRNA (n = 15) (**D**), ERβ siRNA (n = 15) (**E**), siRNA against ERα and ERβ (n = 15) (**F**), and GPER1 siRNA (n = 15) (**G**) on the dendritic areas in Purkinje cells following isoflavone treatment. Bars indicate 50 μm. Data are shown as the mean ± standard error of the mean and represent at least three independent experiments. ^####^
*p*  <  0.0001 indicates statistical significance according to two-way or one-way analysis of variance, followed by the post hoc Tukey test compared with E2 (10 nM). **** *p* < 0.0001 and *** *p* < 0.001 indicate statistical significance according to two-way or one-way analysis of variance, followed by the post hoc Tukey test.

**Figure 3 ijms-24-09011-f003:**
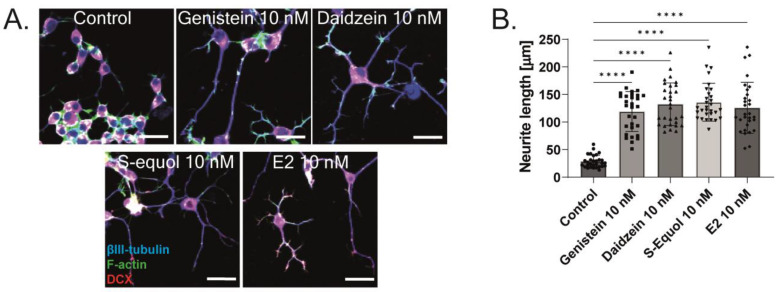
ERα affects isoflavone-induced neurite outgrowth in neuronal-derived Neuro-2A cells. Cells were induced to differentiate through serum starvation for 1–3 days, followed by immunofluorescence analysis with β-tubulin III (blue), doublecortin (red), and F-actin (green). (**A**) Representative photomicrographs showing the effects of isoflavones on the differentiation of Neuro-2A cells. (**B**) Changes in neurite lengths after exposure to isoflavones (n = 30). Quantitative analysis of the effect of isoflavones after exposure to ICI 182,780 (**C**), G15 (**D**) ERa siRNA (**E**), ERb siRNA (**F**), ERα and ERβ siRNA (**G**), and GPER1 siRNA (**H**) on neurite length in Neuro-2A cells (n = 30). Bars indicate 50 μm. Data are shown as the mean ± standard error of the mean and represent at least three independent experiments. ^####^
*p*  <  0.0001 indicates statistical significance according to two-way or one-way analysis of variance, followed by the post hoc Tukey test compared to E2 (10 nM). **** *p* < 0.0001, *** *p* < 0.001, ** *p* < 0.01 and * *p* < 0.05 indicate statistical significance according to two-way or one-way analysis of variance, followed by the post hoc Tukey test.

**Figure 4 ijms-24-09011-f004:**
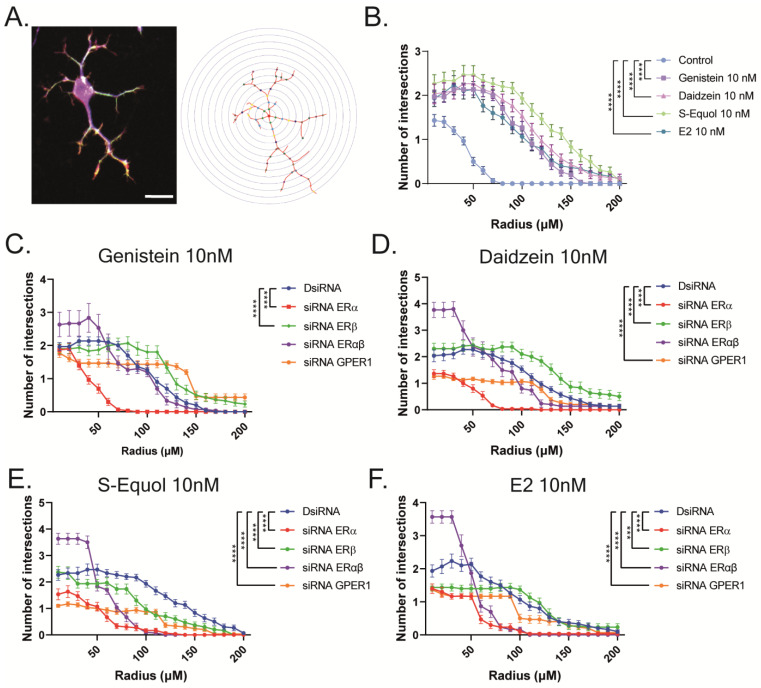
Isoflavone-induced morphological changes in neuronal-derived Neuro-2A cells through the modulation of ERα. Cells were induced to differentiate through serum starvation for 1–3 days, followed by immunofluorescence analysis with β-tubulin III (blue), doublecortin (red), and F-actin (green). (**A**) Diagram of Sholl analysis. (**B**) Quantitative analysis of the effect of isoflavones on the number of neurite intersections at each concentric circle by Sholl analysis. Effects of genistein (**C**), daidzein (**D**), S-equol (**E**), or E2 (**F**) on the number of neurite intersections in Neuro-2A cells after the exposure of DsiRNA, siRNA against ERα, ERβ, or GPER1 (n = 30). (**G**) Representative images of blots for ERα, ERβ, GPER1, or GAPDH levels in response to isoflavone exposure in Neuro-2A cells. Quantitative analysis of the effect of isoflavones on ERα (**H**), ERβ (**I**), and GPER1 protein expression levels (**J**) (n = 6). Bars indicate 50 μm. Data are shown as the mean ± standard error of the mean and represent at least three independent experiments. **** *p* < 0.0001 and *** *p* < 0.001 indicate statistical significance according to two-way or one-way analysis of variance, followed by the post hoc Tukey test.

**Figure 5 ijms-24-09011-f005:**
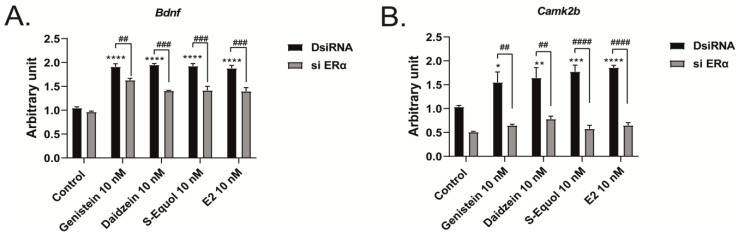
Effects of isoflavones on mRNA expression levels after the knockdown of ERα. Cells were induced to differentiate through serum starvation for 1–3 days, followed by RNA extraction, and continued with qRT-PCR. Changes in *Bdnf* (**A**), *Camk2b* (**B**), *Rbfox3* (**C**), *Tubb3* (**D**), *Syn1* (**E**), *Dlg4* (**F**), *Syp* (**G**), and *Map2* (**H**) mRNA expression levels in Neuro-2A cells after ERα siRNA transfection (n = 9). Data are shown as the mean ± standard error of the mean and repeated three times, using independent RNA preparations to confirm the consistency of the results. mRNA levels were normalized against *Gapdh*. ^####^
*p*  <  0.0001, ^###^
*p* < 0.001, ^##^
*p* < 0.01, and ^#^
*p* < 0.05 indicate statistical significance according to two-way or one-way analysis of variance, followed by the post hoc Tukey test. **** *p* < 0.0001, *** *p* < 0.001, ** *p* < 0.01, and * *p* < 0.05 indicate statistical significance according to two-way or one-way analysis of variance, followed by the post hoc Tukey test compared to E2 (10 nM).

**Figure 6 ijms-24-09011-f006:**
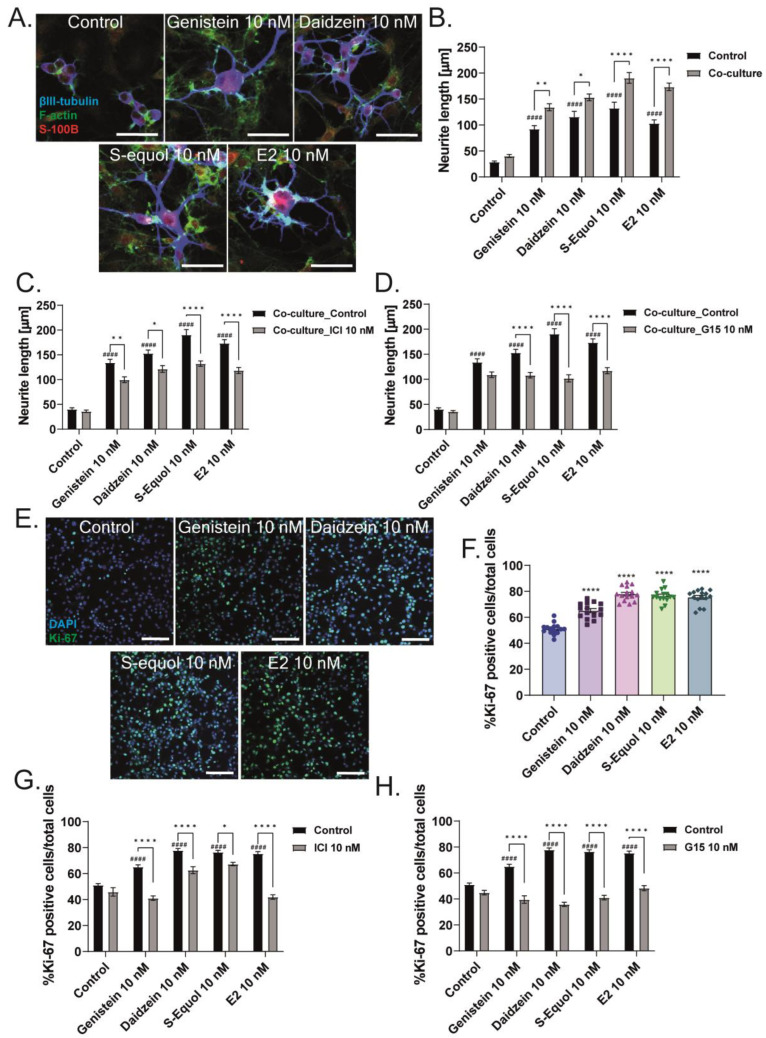
Isoflavones affect neuritogenesis through enhanced neuron–glia interaction and glia proliferation. Astrocyte-enriched cultures were plated until 70–80% confluence, and then Neuro-2A cells were added to the plate. Co-cultures were performed for 1–3 days with serum starvation to induce differentiation. Cells were fixed and immunostained with β-tubulin III (blue), F-actin (green), and S100B (astrocyte marker) (red). (**A**) Representative photomicrographs showing the effects of isoflavones on the differentiation of Neuro-2A cells in the co-cultures of astrocytes and Neuro-2A cells. Quantitative analysis of the effect of isoflavones (**B**) and co-exposure of isoflavones and ICI 182,780 (**C**) or G15 (**D**) on the neurite length of Neuro-2A cells in the co-cultures of astrocytes and Neuro-2A (n = 30). (**E**) Representative photomicrographs showing the effects of isoflavones in the proliferation of astrocytes. Astrocytes were cultured for 3–7 days and then immunostained with Ki-67 as a proliferation marker (green) and DAPI (blue). Quantitative analysis of the effect of isoflavones (**F**) and co-exposure of isoflavones and ICI 182,780 (**G**) or G15 (**H**) on the percentage of Ki-67-positive cells among the astrocytes (n = 15). Bars indicate 50 μm. Data are shown as the mean ± standard error of the mean and represent from at least three independent experiments. ^####^
*p*  <  0.0001 indicates statistical significance according to two-way or one-way analysis of variance, followed by the post hoc Tukey test compared to E2 (10 nM). **** *p* < 0.0001, ** *p* < 0.01 and * *p* < 0.05 indicate statistical significance according to two-way or one-way analysis of variance, followed by the post hoc Tukey test.

## Data Availability

All study data are included in the article and/or Appendix A. All materials are available from the corresponding authors upon request.

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
