# Peer review of "Isoflavones Mediate Dendritogenesis Mainly through Estrogen Receptor α"

_ijms, 2023, doi:10.3390/ijms24109011_

Round 1
Reviewer 1 Report
I read the manuscript with a great interest. The authors evaluated the effects of isoflavones using mouse primary cerebellar culture, astrocyte-enriched culture, Neuro-2A clonal cells, and co-culture with neurons and astrocytes and for studying the specific molecular mechanism, they used Neuro-2A clonal cells. The results demonstrated that ERα is essetnial for isoflavone-induced neuritogenesis. GPER1 signaling is also crucial for astrocyte proliferation and astrocyte–neuron interaction, which may lead to isoflavone-induced neuritogenesis.
The authors used appropriate methods. The results section written in clear and understandable way. All figures ae of high quality and include necessary information. The additional files, including gel images are of high quality.
I definetely recommend this manuscript for publication.
Author Response
I read the manuscript with a great interest. The authors evaluated the effects of isoflavones using mouse primary cerebellar culture, astrocyte-enriched culture, Neuro-2A clonal cells, and co-culture with neurons and astrocytes and for studying the specific molecular mechanism, they used Neuro-2A clonal cells. The results demonstrated that ERα is essetnial for isoflavone-induced neuritogenesis. GPER1 signaling is also crucial for astrocyte proliferation and astrocyte–neuron interaction, which may lead to isoflavone-induced neuritogenesis.
The authors used appropriate methods. The results section written in clear and understandable way. All figures ae of high quality and include necessary information. The additional files, including gel images are of high quality.
I definitely recommend this manuscript for publication.
Response:
Thank you for sharing your positive feedback on the manuscript. It's great to hear that you found the study interesting and well-conducted. It's also great to hear that the figures and additional files are of high quality. We greatly appreciate your feedback.
Reviewer 2 Report
The manuscript of Ariyani and their co-authors is well-written and dedicated to the properties of a few flavonoids in the mediation of dendritogenesis. The data presented in the manuscript is significant and supports the conclusions. There is just one comment that authors should consider before the paper will be accepted.
What rodent diet was used for mice before and during pregnancy? Was it oy-bean-based? What content of isoflavones was in the diet? The content of soy in laboratory animal diets usually varies and may significantly affect the results of the experiment (doi.org/10.3390/nu13103599, doi.org/10.30802/AALAS-JAALAS-18-000129). Thus, the brand of the diet and isoflavones content should be disclosed in Methods, and the potential effect of dietary isoflavones in brain cells should be discussed.
Author Response
We appreciate greatly your comments and suggestions. We have made several alterations to improve our manuscript according to your comments. In the revised version of the manuscript, changed sentences were marked in red. Here are the responses in detail:
The manuscript of Ariyani and their co-authors is well-written and dedicated to the properties of a few flavonoids in the mediation of dendritogenesis. The data presented in the manuscript is significant and supports the conclusions. There is just one comment that authors should consider before the paper will be accepted.
What rodent diet was used for mice before and during pregnancy? Was it soy-bean-based? What content of isoflavones was in the diet? The content of soy in laboratory animal diets usually varies and may significantly affect the results of the experiment (doi.org/10.3390/nu13103599, doi.org/10.30802/AALAS-JAALAS-18-000129). Thus, the brand of the diet and isoflavones content should be disclosed in Methods, and the potential effect of dietary isoflavones in brain cells should be discussed.
Response:
In this study, we used the CE-2 standard rodent diet from CLEA Japan, Inc., which contains soybean and soybean oil. We agree with the reviewer that dietary isoflavones during pregnancy may affect the brain cells since brain development starts in the fetal period. Therefore, we also conducted a study to examine the effect of a maternal standard rodent diet and a soybean-free diet on brain development and behavior. However, we conducted this study in collaboration with other institution. Due to the conflict of interest, we cannot show the data at present. We also measured the isoflavones concentrations in the CE-2 diet, and it was less than those of dietary isoflavones in humans. We plan to submit our findings as a separate manuscript, soon. In the revised manuscript, we included the detail of the mice's diet in the Material and Methods section. We also added the potential effect of dietary isoflavones in the discussion section. We greatly appreciate your kind understanding.
Reviewer 3 Report
In the present study, Ariyani and colleagues determine the effects of isoflavones on dendritic arborization in different cell-types. The present study shows that isoflavones exert their effect on dendritic morphology through ERs and GPER1 with ERα playing a greater role in affecting these changes.
Below are some comments that need to be addressed before the manuscript can be considered for publication:
In Figure 1 C, and Figure 2 B, the quantified data presented is “relative dendritic area” but the methods section (lines 395-397) does not describe how this quantification was done. Which ImageJ macro was used for quantification?
In lines 202-203, the authors speculate that the increased neurite branches observed with knockdown of ERα and ERβ could be due to the binding of isoflavones to GPER1. However, no follow up experiments were performed to determine that. It would be interesting to see how dendritic arborization is affected by the simultaneous knockdown of ERαβ and GPER1
Lines 204-209: While isoflavones increased the levels of ERα, knockdown of ERβ as well as GPER1 affects number of branches and neurite length. It would be interesting to see whether genes regulated by ERβ and GPER1 are also affected by exposure to isoflavones.
In lines 238-240, the authors mention that isoflavones lead to rearrangement of F-actin. Were any changes in the cytoskeleton observed in these neurons due to isoflavones?
In lines 394-395,418-419, 439-440, 457-458, imaging parameters (lens objective, step size, resolution, etc.) should be mentioned.
Minor comments:
Figure 2 was difficult to read as portion of it was cut off.
For the original Western blot image, each lane should be specified.
For figure 6, the scale bar is not mentioned in the figure legends.
Author Response
We appreciate greatly your comments and suggestions. We have made several alterations to improve our manuscript according to your comments. In the revised version of the manuscript, changed sentences were marked in red. Here are the responses in detail:
In the present study, Ariyani and colleagues determine the effects of isoflavones on dendritic arborization in different cell-types. The present study shows that isoflavones exert their effect on dendritic morphology through ERs and GPER1 with ERα playing a greater role in affecting these changes.
Below are some comments that need to be addressed before the manuscript can be considered for publication:
Point 1:
In Figure 1 C, and Figure 2 B, the quantified data presented is “relative dendritic area” but the methods section (lines 395-397) does not describe how this quantification was done. Which ImageJ macro was used for quantification?
Response 1:
We did not use the ImageJ macro to quantify the dendritic area. We manually traced the dendritic area of Purkinje cells with ImageJ Freehand selection tools. We added the explanation in the Material and Methods section 4.2 in the revised manuscript.
Point 2:
In lines 202-203, the authors speculate that the increased neurite branches observed with knockdown of ERα and ERβ could be due to the binding of isoflavones to GPER1. However, no follow up experiments were performed to determine that. It would be interesting to see how dendritic arborization is affected by the simultaneous knockdown of ERαβ and GPER1
Response 2:
We have tried to knockdown ERαβ and GPER1 in both primary cultures of mouse cerebellar cells and Neuro-2A cells. However, additional knockdown of GPER1 needed to increase the amount of Lipofectamines RNAimax for the siRNA transfection, which led to increased toxicity in the cells. Therefore, we tried to co-exposure of both ICI 182,780 (ER antagonist) and G15 (GPER1 antagonist) in the cultures. The isoflavone-induced dendritic outgrowth was significantly reduced by this co-exposure, almost similar to the levels of the control groups. However, it was difficult to draw clear conclusions from these results, as the contribution of each signaling pathway through each receptor could not be determined in such experiments. Therefore, we did not include the data in this manuscript. We greatly appreciate your understanding.
Point 3:
Lines 204-209: While isoflavones increased the levels of ERα, knockdown of ERβ as well as GPER1 affects number of branches and neurite length. It would be interesting to see whether genes regulated by ERβ and GPER1 are also affected by exposure to isoflavones.
Response 3:
We also examined the mRNA levels of Bdnf, Camk2b, Rbfox3, Tubb3, Syn1, Dlg4, and Syp after the knockdown of ERβ or GPER1. However, we could not find significant differences in all isoflavones compounds. Therefore, we did not include the data. We added the comment in the first paragraph of the discussion section. In the future, we would like to examine the single cells RNA-seq after the knockdown of ERα, ERβ, or GPER1 in the cerebellum during development. Therefore, we could determine the gene expression that is affected by isoflavones exposure. We greatly appreciate your understanding.
Point 4:
In lines 238-240, the authors mention that isoflavones lead to rearrangement of F-actin. Were any changes in the cytoskeleton observed in these neurons due to isoflavones?
Response 4:
In the previous study, we found that isoflavones exposure affects the F-actin rearrangement in astrocytes (https://doi.org/10.3390/ijms20205178; https://doi.org/10.3389/fendo.2020.554941). In this study, we co-stained the co-cultured cells with Phalloidin F-actin, and neuron and astrocyte markers (Fig. 6A). Although we found an increase in filopodia formation after the exposure of isoflavones compared to the control, it was difficult to quantify the change in F-actin levels under the present preparation.
Point 5:
In lines 394-395,418-419, 439-440, 457-458, imaging parameters (lens objective, step size, resolution, etc.) should be mentioned.
Response 5:
We added an explanation regarding the imaging parameter in the Materials and Methods section in the revised manuscript. We appreciate that you pointed it out.
Minor comments:
Point 6:
Figure 2 was difficult to read as portion of it was cut off.
For the original Western blot image, each lane should be specified.
For figure 6, the scale bar is not mentioned in the figure legends.
Response 6:
We replaced the Fig. 2, specified each lane in full-length blot, and added an explanation regarding the scale bar in Fig. 6 figure legends. We appreciate that you pointed it out.
Round 2
Reviewer 3 Report
The authors have satisfactorily addressed the questions and concerns that were raised and the manuscript can be accepted.